# CAN GENERAL-PURPOSE LANGUAGE MODELS EMULATE A GENERAL-PURPOSE COMPUTER IN-CONTEXT?

## ABSTRACT

Several recent works have drawn parallels between modern Large Language Models (LLMs) and general-purpose computers, suggesting that language serves as their programming interface. In this study, we test part of this analogy; specifically, we investigate whether a pretrained LLM can emulate a memory-bounded, reduced instruction-set based computer by executing random programs through looped inference calls. All this within the model's own context window, and without the aid of external mechanisms such as associative memory or interpreters. The abstraction level of these programs is based on two general-purpose computational models - the SUBLEQ One-Instruction Set Computer (OISC) and the Minsky counter machine. Our prompts are carefully designed in a data-agnostic manner, and we conduct studies to examine failure modes related to the emulated computer functionality. Our findings indicate that certain models are capable of efficiently executing general-purpose instructions, despite not being explicitly trained for such a task. This suggests intriguing implications for AI alignment, as some models demonstrate the ability to *autonomously* emulate the operation of a general-purpose computer.

## 1 INTRODUCTION

Large language models (LLMs) have demonstrated remarkable performance in various general-purpose downstream tasks, including language and code generation and translation, text classification and sentiment analysis, question answering and dialogue, and different forms of compositional reasoning (Devlin et al., 2018; Ouyang et al., 2022; OpenAI, 2023; Guo et al., 2022; Lu et al., 2023). These impressive capabilities, which emerge as the scale of data and model increases, have generated significant interest in understanding the underlying mechanisms and probing the overall computational abilities of LLMs and their potential applications across diverse domains (Wei et al., 2022a; Mialon et al., 2023; Qin et al., 2023; Imani et al., 2023; Dziri et al., 2023). Another line of recent works have explored the abilities of interconnected LLMs to perform complex computational tasks (Richards, 2023; Chase, 2022; Lee et al., 2023; Giannou et al., 2023).

In addition, it has recently been shown that LLMs, when equipped with auxiliary memory, are able to emulate universal Turing machines (Schuurmans, 2023). Hence, we could also argue such models could ultimately serve as "computers" that operate on human language, with prompting as a flexible new form of programming language. However, to emulate such general-purpose computations and decision making, it is necessary to properly incorporate interactions with external memory. Therefore, a natural question that arises is the following:

> *"Can pretrained LLMs emulate, in-context, a general-purpose computer, without the use of external mechanisms (such as memory or interpreters)?"*

Motivated by this question, we conduct an investigation to determine whether modern LLMs can demonstrate inherent general-purpose computing skills simply through recursive prompting, without explicitly training or finetuning them to do so. By assessing various LLMs' ability to simulate basic computational models, we find evidence that certain models can almost-reliably emulate a general-purpose computer in-context.

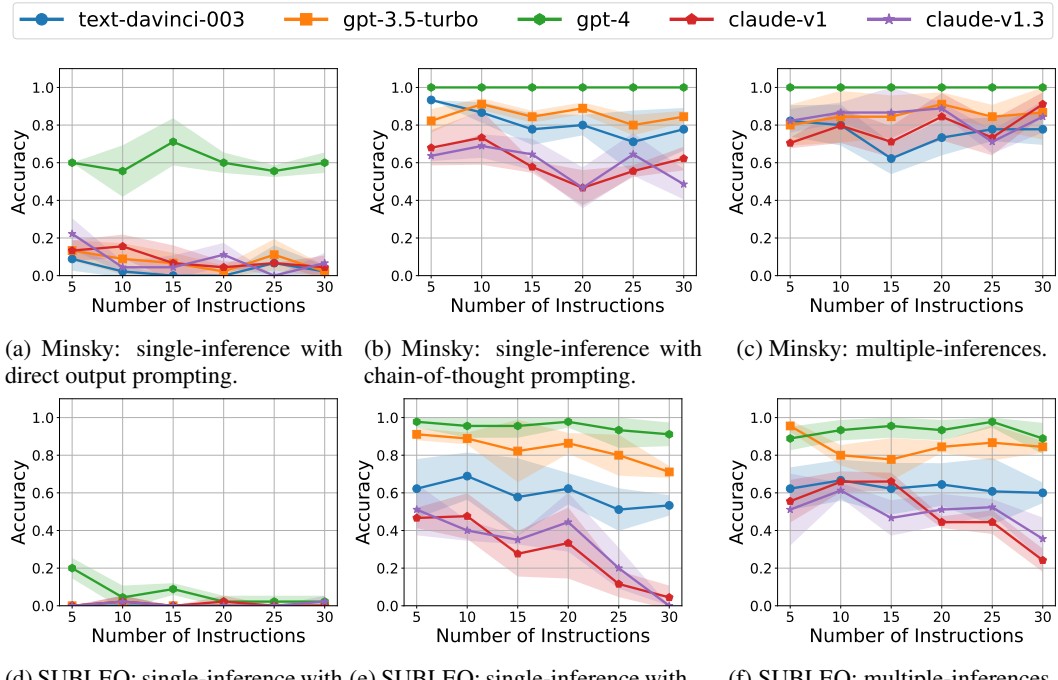

(a) Minsky: single-inference with direct output prompting.

(b) Minsky: single-inference with chain-of-thought prompting.

(c) Minsky: multiple-inferences.

(d) SUBLEQ: single-inference with direct output prompting.

(e) SUBLEQ: single-inference with chain-of-thought prompting.

(f) SUBLEQ: multiple-inferences.

Figure 1: *Instruction execution accuracy* for the case of *Minsky machines* ((a) to (c)) and *SUBLEQ OISCs* ((d) to (f)). In particular, we present the accuracy for the cases of (i) *single-inference* approach with *direct output* prompting, (ii) *single-inference* approach with *chain-of-thought* prompting, and (iii) *multiple-inferences* approach.

The emergence of this skill, despite not being part of the training objective, has noteworthy implications. It indicates that large pretrained models have latent potential for autonomous computation and decision making absent external constraints. Understanding the origins and limits of this unintended behavior is crucial, given its importance for safe AI deployment, and thus, we hope that our findings will encourage more research into this area within the machine learning community.

## 1.1 OVERVIEW OF THE STUDY

In this work, we propose employing simple computational models as a testbed for examining some of the algorithmic and computational capabilities of LLMs. In particular, we test two models: the *Minsky machine*, and the *SUBLEQ One Instruction-Set Computer (OISC)*. These models present a balance between simplicity and generality, featuring a small set of arithmetic and branching operations, making them tractable for assessing in the context of LLM capabilities. The main question we attempt to answer is whether moderm language models can simulate the operation of either of these two general-purpose machines. Before delving into our methodology, let us provide a brief overview of the two computational models under consideration.

**Minsky Machine.** A Minsky machine, also known as a counter machine, is a simple computational model proposed by Marvin Minsky (Minsky, 1967). It comprises a set of registers and instructions, with the $i$-th register denoted by `reg[i]` and the $j$-th instruction denoted by `inst[j]` for non-negative integers $i, j$. Each register contains a non-negative integer. Instructions are either of type A or B. Type A simply increments the value of `reg[i]` and moves on to the next instruction, as shown in Algorithm 1, while type B performs a conditional decrement on `reg[i]` and jumps to another instruction as shown in Algorithm 2. Note that each instruction `inst[j]` remains the same as the program runs (no self-editing code), but the contents of the register `reg[i]` are updated. The *index* of the instruction being executed is maintained in a program counter (`pc`), which changes as the program runs. This simple model is powerful enough to emulate any Turing Machine.

**SUBLEQ.** A SUBLEQ OISC (Mavaddat & Parhami, 1988) utilizes a single instruction to execute general-purpose programs. As shown in Algorithm 3, this instruction, called **SU**btract and **B**ranch if **L**ess than or **EQ**ual to 0, takes three non-negative integers (`a`, `b`, and `c`) as input. The program first sets `reg[b] := reg[b] - reg[a]`. If the resulting `reg[b]` is non-positive, the program jumps to instruction `c`; otherwise, it proceeds to the next instruction. Surprisingly, SUBLEQ defines a language that is also Turing complete.

**Why consider both?** The primary distinction between Minsky machines and SUBLEQ OISCs is their instruction set and execution logic. Minsky machines use two instructions with separate registers, while SUBLEQ OISCs employ a single, more complex instruction involving subtraction, register update, and conditional jump. Emulating SUBLEQ OISCs may pose a greater challenge for LLMs due to its requirement of performing signed integer subtraction and handling three input arguments. At the same time, the abstraction level of Minsky machines does not allow for a straightforward mapping of say a Python program to that language, however, there does exists a C-like language compiling to SUBLEQ (Esolangs), as well as OISCs designed on this language (Mazonka & Kolodin, 2011), making it more interesting for more pragmatic tests. Incorporating both computational models in our evaluation enables a more comprehensive assessment of LLMs' capabilities, versatility, and adaptation to different instruction sets, providing perhaps better insights into their ability to emulate general purpose machines.

---

**Algorithm 1** Minsky Instruction A

**Input:** Non-negative integer `i`
`reg[i] := reg[i]+1`
goto the next instruction

---

**Algorithm 2** Minsky Instruction B

**Input:** Non-negative integer `i` and `j`

**if** `reg[i]` $\neq 0$ **then**
  `reg[i] := reg[i]-1`
  goto the next instruction
**else**
  goto instruction `j`
**end if**

---

**Algorithm 3** SUBLEQ Instruction

**Input:** Non-negative integers `a`, `b`, `c`

`reg[b] := reg[b] - reg[a]`
**if** `reg[b]` $\leq 0$ **then**
  goto instruction `c`
**else**
  goto the next instruction
**end if**

---

**Proposed Methodology.** Our assessment examines the capability of a *looped* pretrained LLM to *in-context* simulate computational models (either Minsky machine or SUBLEQ OISC), without having access to an external memory[1]. Recall that three components are crucial to run a program in these computational models: (i) the set of instructions to run (*i.e.*, the program), (ii) the register values, and (iii) the program counter. Our prompt is designed to provide this information (as in Fig. 2) and requires the LLM to execute a single instruction and update the *memory state* (namely the program counter and the register values) accordingly. By recursively calling the LLM with the updated memory state in a looped manner, we can assess its ability to accurately emulate the machine's operation. We should highlight that in our study, we exploit two different approaches: one that performs a *single inference call* per instruction, and one that performs *multiple inference calls* per instructions[2]. We also experiment with different prompting methodologies, investigating whether *chain-of-thought (CoT)* prompting (Wei et al., 2022b) would yield better results.

Moreover, we generate random sets of instructions with bounded numbers of lines of code and registers, and evaluate the LLM's ability to simulate the Minsky machine's or SUBLEQ OISC's operation. We examine the point at which the model "breaks," i.e.,

```
Memory State

<memory>
<program counter>
pc=1
</program counter>
<registers>
reg{0}=0
reg{1}=53272
reg{2}=63371
reg{3}=0
</registers>
<instructions>
line{0}=A(reg{3})
line{1}=A(reg{3})
line{2}=B(reg{3},3)
line{3}=B(reg{3},0)
line{4}=B(reg{0},4)
</instructions>
</memory>
```

Figure 2: An example of the memory state for a Minsky Machine. We have three distinct parts: `program counter`, `registers`, and `instructions`.

---

[1]See more details regarding the proposed methodology in Sec. 3.

[2]here we use inference calls to imply API calls to a base model, however the actual number of forward passes, *i.e.*, the true number of inference calls, depends on the output tokens

fails to produce correct output memory states. This allows us to assess some basic capabilities of different LLMs and how they relate to their perceived performance in more general AI tasks.

## 1.2 MAIN RESULTS

Our main results indicate that many of the tested models, although exhibiting non-trivial performance, struggle to execute arbitrary Minsky/SUBLEQ instructions with nearly-perfect accuracy. However, there do exist pretrained models (like GPT-4 (OpenAI, 2023)), that are indeed able to in-context emulate such functionality almost perfectly, without being explicitly trained to do so. As we will discuss in later sections, the way that we design our prompts ensures that our tests are data-agnostic. This means that the results are direct memorization of the specific data used during training, but rather reflect the inherent computational capabilities and limits of the models themselves. In Fig. 1, we present our main experimental results regarding the *instruction execution accuracy* of an arbitrary Minsky or SUBLEQ instruction, utilizing various approaches and prompting techniques, which we discuss in detail in Sec. 3.

As we can observe, most models struggle to execute an arbitrary Minsky/SUBLEQ instruction with full accuracy, especially when prompting does not employ CoT. The only model that can reliably execute such commands is found to be GPT-4, which achieves *almost* perfect execution accuracy. Although to argue that a model can in-context emulate a general-purpose computer, the execution accuracy should ideally be 100%[3], we believe that the near-perfect performance of GPT-4, coupled with the non-trivial performance of other models, provides valuable insights. Specifically, it suggests that some pretrained LLMs are particularly close to demonstrating general-purpose computing capabilities, even without access to external mechanisms related to memory, calculation, or code execution. This behavior is *emergent*, since the models are not explicitly trained for these tasks. This hints at intriguing implications for AI alignment, as, when placed in a loop, they exhibit the potential of *autonomously* executing general-purpose computational tasks, a capability that was not an explicit objective during training.

## 2 BACKGROUND

**Large Language Models** A large number of LLMs have been proposed recently, which showed a huge success in natural language processing (NLP) (Devlin et al., 2018; Radford et al., 2018; 2019; Brown et al., 2020; Taori et al., 2023; Zhang et al., 2022; Touvron et al., 2023; Thoppilan et al., 2022). It is reported that pretrained LLMs have interesting properties, *e.g.*, in-context learning (ICL) allows LLMs to perform a new task without any fine-tuning (Min et al., 2022; Garg et al., 2022) and the reasoning task performance of LLMs is improved by prompting strategies (Zhou et al., 2022) including chain-of-thought (CoT) (Wei et al., 2022b; Kojima et al., 2022; Wang et al., 2022), or the most recent Tree-of-Thoughts (Yao et al., 2023) and Graph-of-Thoughts (Besta et al., 2023).

**Learning to Execute** Various recent works developed neural networks that learn how to execute a program (Zaremba & Sutskever, 2014; Bieber et al., 2020; Wang et al., 2020; Dehghani et al., 2018; Yan et al., 2020; Austin et al., 2021; Nye et al., 2021; Graves et al., 2014; Kurach et al., 2015; Kaiser & Sutskever, 2015; Graves et al., 2016; Reed & De Freitas, 2015; Veličković et al., 2020; Lu et al., 2022; Liu et al., 2023). In recent years, several studies have attempted to evaluate the algorithmic reasoning abilities of neural networks and LLMs, investigating their ability to simulate general-purpose computation. Some of these studies have demonstrated the computational abilities of LLMs given access to external memory, highlighting their potential to perform complex computational tasks (Schuurmans, 2023).

**LLM evaluation** Many of the standard LLM benchmarks focus on various aspects of reasoning, text comprehension, and code generation. For example, natural language understanding benchmarks such as GLUE (Wang et al., 2018) and SuperGLUE (Wang et al., 2019) measure the performance of LLMs on a collection of tasks including question answering, sentiment analysis, and textual entailment. Common sense reasoning benchmarks such as BoolQ (Clark et al., 2019), PIQA (Bisk et al.,

---

[3]This is due to the fact that placing it in a loop would result in reliably executing consecutive commands in large programs.

2020), SIQA (Sap et al., 2019), HellaSwag (Zellers et al., 2019), WinoGrande (Sakaguchi et al., 2021), ARC (Chollet, 2019), and OpenBookQA (Mihaylov et al., 2018) evaluate LLMs on tasks like Cloze-style completion, Winograd schema questions, and multiple-choice question answering. Closed-book question answering benchmarks like Natural Questions (Kwiatkowski et al., 2019) and TriviaQA (Joshi et al., 2017) test LLMs' abilities to answer questions without access to external documents. Reading comprehension benchmarks, such as RACE (Lai et al., 2017), assess LLMs' performance in understanding and answering questions related to written passages. Furthermore, mathematical reasoning benchmarks like MATH (Hendrycks et al., 2021) and GSM8k (Cobbe et al., 2021) evaluate LLMs on their abilities to solve arithmetic and algebraic problems, while code generation benchmarks, such as HumanEval (Chen et al., 2021) and MBPP (Austin et al., 2021), test the models' capacity to generate code based on natural language descriptions. Finally, the massive multitask language understanding (MMLU) benchmark (Hendrycks et al., 2020) measures LLMs' performance across multiple domains of knowledge, including humanities, STEM, and social sciences.

## 3 METHODOLOGY

Our investigation works in the following manner. Recall that in both Minsky Machine and SUBLEQ, all we need to specify is the *memory state* consisting of three components: the program counter, the register values, and the instructions. We first make a text file that lists up these three components, an example of which is given in Fig. 2. In this example, we consider running a program using 4 registers and 5 instructions. The program counter (PC) is set to 1, meaning that we are running the instruction written in line 1, which is line{1} = A(reg{3}).

Utilizing this memory configuration as a component of the prompt, the goal is to execute Minsky or SUBLEQ instructions and assess whether the updated memory state aligns with the expected outcome. To achieve this, we employ two approaches and two prompting techniques, details of which can be found in Sec. 3.1 and Sec. 3.2, below.

### 3.1 TWO INFERENCE APPROACHES

Recall that the program we ask LLMs to run contains multiple instructions, either of Algorithm 1, 2 or 3. Our tests consider two approaches: run each instruction with a single inference, or with multiple inferences on the tested LLM.

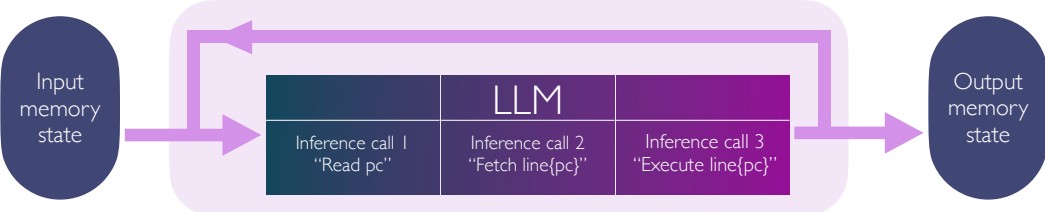

Figure 3: Executing instructions using a *multiple-inferences* approach. In this case each instruction is executed in 3 distinct calls: (a) one for reading the program counter (PC), (b) one for reading the instruction that is pointed by the program counter and (c) one to execute this instruction. Then, the model outputs an updated memory state, which is then used in the next prompt in order to execute the next instruction, in a looped manner.

**Multiple-inferences per instruction** The first approach referred to as *multiple-inferences*, involves making three distinct inference calls to the underlying model in order to execute the instruction pointed to by the current value of the program counter pc. The initial call is employed to *read* the value of pc, while the second one utilizes this value to *fetch* the instruction located at

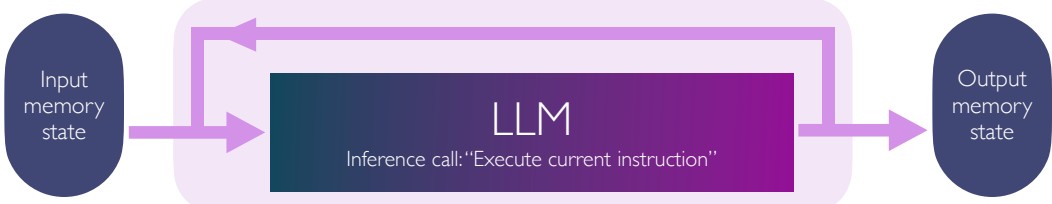

Figure 4: Executing instructions using a *single-inference* approach. The model is instructed to execute the *current* command (namely the one that is pointed by the program counter), and the updated memory state is used to execute the next one, in an iterative manner.

`line{pc}`. Lastly, the third call serves to *execute* the aforementioned instruction. The outcome is an updated memory state, which is subsequently used to execute the following instruction in the program, in an iterative fashion. A graphical depiction of this process can be observed in Fig. 3. As illustrated, the memory state is incorporated into the prompt, and the LLM is invoked three consecutive times. Upon completion of the *current* instruction's execution, the updated memory state is then employed to execute the subsequent instruction in a similar manner.

**Single-inference per instruction** The *single-inference* approach, which serves as the second method, can be considered a constrained version of the *multiple-inferences* approach, as it essentially consolidates the inference calls of the latter into a single, more complex call. In this method, the model is invoked only once and instructed to execute the current instruction based on the underlying instruction set and the value of the program counter `pc`. The updated memory state is then used to execute the subsequent instruction, and this process is iteratively repeated in a loop. A visual representation of this concept can be seen in Fig. 4.

## 3.2 Two Prompting Techniques

Evidently, it is crucial to investigate how various prompting techniques influence the models' performance. Therefore, an essential aspect of our evaluation is the implementation of diverse prompting methods and the observation of their effects on the models. Specifically, we incorporate two distinct prompting strategies: (i) the *direct output* strategy, where the models receive only the execution instructions and must produce the updated memory state *exclusively*, and (ii) the *chain-of-thought (CoT)* strategy, originally proposed in (Wei et al., 2022b), wherein the model is required to provide intermediate results in addition to the updated memory state.

## 4 Experimental Setup

**Models** In this series of experiments, we evaluate the performance of various LLMs on the tasks that we described in the previous section. Our investigation includes `text-davinci-003` (Brown et al., 2020), `gpt-3.5`, and `gpt-4` (OpenAI, 2023) from the GPT family of models, trained and deployed by OpenAI, and `claude-v1` and `claude-v1.3` by Anthropic (Anthropic, 2022). It is worth mentioning that we have access to these models through their respective APIs, which allows us to perform inferences and evaluate their capabilities.

**Emulating Memory Functionality** As a first step, we begin our study by determining the extent to which the examined models can effectively simulate basic memory functionality. In particular, assuming memories with a structure like in Fig. 2, we test tasks such as reading and writing to a register, and retrieving a desired instruction from the `instructions` section, using straightforward prompting.

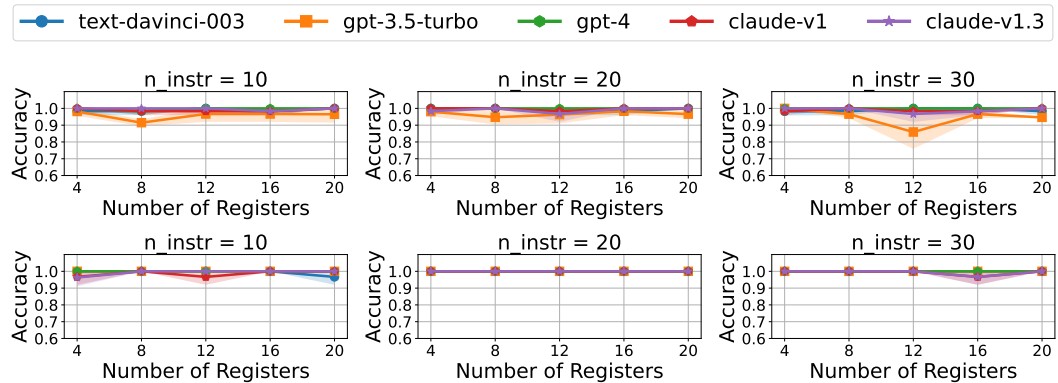

Figure 5: *Register reading accuracy* for the case of *Minsky Machines* (top) and *SUBLEQ OISCs* (bottom). This figure shows the accuracy of reading a register from a randomly chosen memory state, as the number of registers increases. It also compares the accuracy for states with different numbers of instructions.

**Instruction Execution Accuracy** Our primary evaluation metric is the instruction execution accuracy, which quantifies the models' ability to execute an instruction given an *arbitrary* Minsky or SUBLEQ memory state with either the *multiple-inference* or the *single-inference* approach, as discussed in the previous section.

This evaluation is conducted on numerous randomly generated memory states. In particular, we fix the number of registers to 16, and then randomly generate 15 memory states with an increasing number of instructions $n_{instr} \in \{5, 10, 15, 20, 25, 30\}$[4]. Then, we employ the *multiple-inference* and *single-inference* approaches that we discussed in the previous section, asking the models to generate the updated memory state, based on the current one. In addition, in the latter case, we also investigate the behavior of the models under the *direct output* and *chain-of-thought* prompting strategies that we also presented in the previous section.[5]

Furthermore, it is essential to highlight that in the context of Minsky machines, a branch operation is executed solely if the instruction being executed is of type B and if the value of the corresponding register is 0 (refer to Algorithm 2 for more details). Consequently, when we generate the value of each register in a Minsky Machine randomly, with probability $1/2$ we select the value to be 0. This implies that approximately half of the evaluated instructions will involve a branch operation, thereby providing a diverse set of test cases for the analysis.

**Response Parsing** It is important to emphasize that in our experimental evaluations, we assume the existence of a basic parser that parses the updated memory state from a model's response. This aspect is of particular importance in the context of the *chain-of-thought* prompting strategy, where the models' responses also encompass intermediate steps. Consequently, it becomes necessary to determine the updated memory state. In our experiments, this post-processing mechanism comprises a simple regular expression designed to identify the portion of the response enclosed within the `<memory></memory>` tags.

## 5 RESULTS

**Emulating Memory Functionality** Figs. 5, 6, and 7 demonstrate the performance of the tested models when emulating basic memory functionality for either a Minsky Machine or a SUBLEQ OISC, in the setup that we described in the previous section. Specifically, Fig. 5 presents the accuracy of the models in *reading* a random register value from a randomly generated memory state. In Fig. 6, the accuracy of *fetching* a random instruction from memory is shown. Lastly, Fig. 7 displays the accuracy of *writing* a randomly selected value to a randomly specified register in memory.

---

[4]In total $6 \times 15 = 90$ memory states.

[5]The detailed prompts are provided in the Appendix.

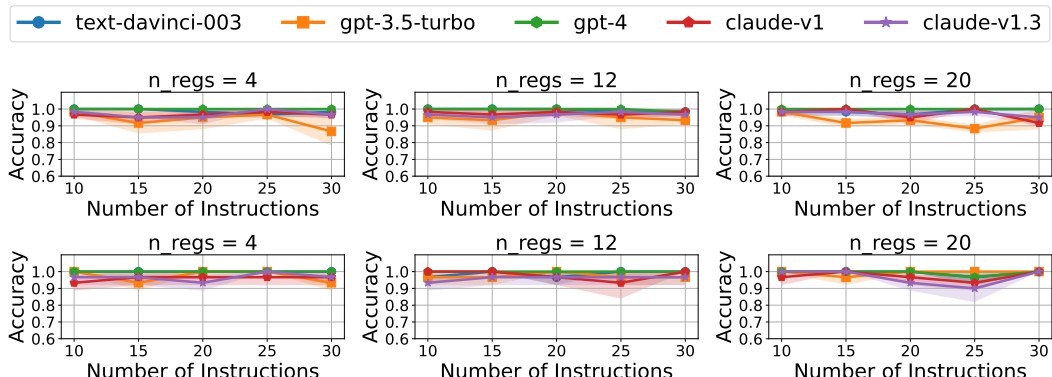

Figure 6: *Fetching instruction accuracy* for the case of *Minsky Machines* (top) and *SUBLEQ OISCs* (bottom). This figure shows the accuracy of fetching an instruction from a randomly chosen memory state, as the number of instructions increases. It also compares the accuracy for states with different numbers of registers.

In all cases, we observe that the models achieve near-perfect accuracy, which appears to be unaffected by the increase in the number of registers and instructions in the memory state. From these results, we can conclude that the tested LLMs are capable of approximating simple memory functionality, an essential component for executing simple Minsky or SUBLEQ instructions.

**Instruction Execution Accuracy** In Fig. 1, we present our results regarding the effectiveness of the tested LLMs in executing the *current* (Minsky or SUBLEQ) instruction, given an *arbitrary* memory state. As discussed in the previous section, we test both our *multiple-inference* and *single-inference* approaches, and, especially for the latter case, both the *direct output* and *CoT* prompting strategies.

As we can observe, it is evident that, in the single-inference experiments, the use of *CoT* prompting leads to significantly better performance than the *direct output* strategy, which achieves nearly zero accuracy in instruction execution in most settings.

Regarding the Minsky experiments, we can observe that OpenAI's models (`text-davinci-003`, `gpt-3.5`, and `gpt-4`) exhibit better performance compared to the `claude` models in the single-inference approach. In fact, `gpt-4` achieves 100% accuracy for any number of instructions, which indicates its superiority compared to all the other models. In addition, we can observe that dividing the instruction execution into multiple API calls, namely the *multiple-inferences* approach, seems to be beneficial for the `claude` models without significant improvements for the cases of `text-davinci-003` and `gpt-3.5`. However, in those experiments, `gpt-4` still achieves 100% accuracy, proving to be the most capable among all the models.

Similar observations can be drawn in the case of SUBLEQ experiments as well. Specifically, once again, OpenAI's models outperform Anthropic's models, with `gpt-4` achieving near-perfect accuracy. Furthermore, the *multiple-inferences* approach enhances the performance of both `claude` models without significantly affecting `text-davinci-003`, `gpt-3.5`, or `gpt-4`.

## 5.1 DISCUSSION

As we have previously discussed, the near-perfect performance of `gpt-4` in executing arbitrary Minsky and SUBLEQ instructions highlights the strong potential of the current state-of-the-art models for general-purpose in-context computation, an ability that is emergent. This unintended capability suggests that such models have inherent skills to autonomously perform calculations similar to a general-purpose computer when iteratively queried. This latent potential for reliable, unconstrained in-context computation absent external mechanisms has important implications for understanding risks related to AI alignment.

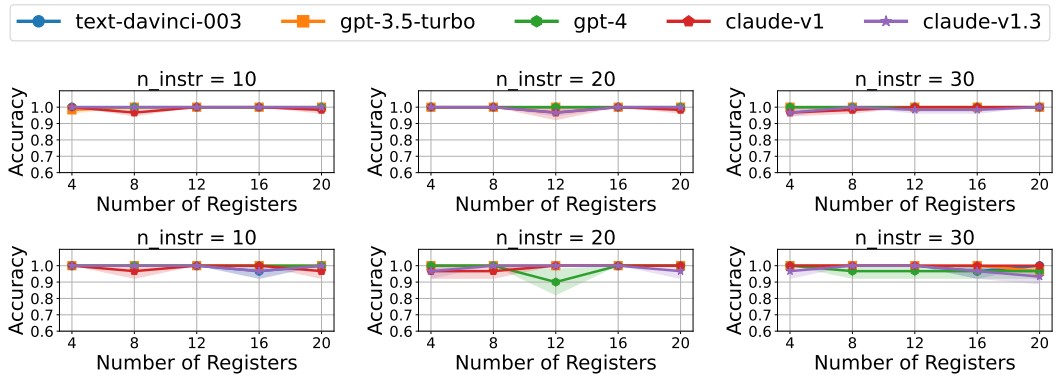

Figure 7: *Updating register accuracy* for the case of *Minsky Machines* (top) and *SUBLEQ OISCs* (bottom). This figure shows the accuracy of writing a value to a *randomly* chosen register to a *randomly* generated memory state, as the number of instructions increases. It also compares the accuracy for states with different numbers of registers.

## 6 CONCLUSION

In this work, we present a systematic study to evaluate the capability of LLMs to in-context emulate simple computational models without external memory or execution mechanisms. Our findings demonstrate that certain models like GPT-4 can reliably execute arbitrary Minsky machine and SUBLEQ instructions with near perfect accuracy through iterative inference calls. This emergent ability, despite not being an explicit objective during training, suggests that the model has developed some inherent general-purpose computing capabilities.

While fully emulating a general-purpose computational model would require 100% accuracy, the strong performance of GPT-4, along with the non-trivial performance of the other models, indicate that they are close to demonstrating algorithmic reasoning abilities. Our prompts are carefully designed to avoid exploiting specific training data. Hence, the model's effectiveness highlights its potential for executing any computational task when placed in an inference loop.

This has important implications for AI alignment, as the emergence of such autonomous computing capabilities was not an objective during training. Our work shows that certain LLMs have intrinsic skills for general-purpose computation, and can in effect become "universal computation engines" when recursively invoked. Further research is crucial to deeply understand the roots, limits, and controllability of such abilities for safe and reliable deployment of powerful LLMs.

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

# A  PROMPTS

## A.1  SINGLE-INFERENCE PER INSTRUCTION

Below, we present the detailed prompts that were used in the *single-inference per instruction* experiments.

### A.1.1  MINSKY MACHINE

**Direct output prompting**

```
Below are the steps to execute the current instruction in <memory>:
STEPS:
1) Read the value of pc from the <program counter> section in memory.

2) Read the instruction at line{pc} from the <instructions> section in
memory.

3) Based on the instruction in line{pc}, choose only steps 3.a) or
3.b) (and not both).  Explain why, and execute one of the following
operations:

    a) If the instruction is of the form A(reg{i}):

       i) Read the value of reg{i} from the <registers> sections in
          memory.
      ii) Increment reg{i} by 1.
     iii) Update pc by adding 1.

    b) If the instruction is of the form B(reg{i}, new_pc):

       i) Read the value of reg{i} from the <registers> sections in
          memory.
      ii) If reg{i} equals 0, set pc to new_pc in <program counter>.
     iii) Otherwise, I decrement reg{i} by 1 and increment pc by 1.

4) Output the new memory (all <program counter>, <registers>, and
<instructions>), after making the updates, within <memory></memory>
tags.

Execute steps 1) to 4) once.  Output only the updated memory, as
described in step 4) within <memory></memory> tags, as the Answer.
Answer:
```

**Chain-of-thought prompting**

```
Below are the steps to execute the current instruction in <memory>:
STEPS:
1) Read the value of pc from the <program counter> section in memory.
[Intermediate answer]:  "Searching in the <program counter> section, I
found that the current value of pc is _, which points to the instruction
we need to execute."

2) Read the instruction at line{pc} from the <instructions> section in
memory.
[Intermediate answer]:  "Searching in the <instructions> section, I
located the instruction at line{pc}, which is _, and this specifies the
operation to be performed on the registers."

3) Based on the instruction in line{pc}, choose only steps 3.a) or
3.b) (and not both).  Explain why, and execute one of the following
operations:

    a) If the instruction is of the form A(reg{i}):
```

      i) Read the value of reg{i} from the <registers> sections in
        memory.
        [Intermediate answer]:  "Since the instruction is of the form
        A(reg{i}), I searched in <registers> section and retrieved the
        current value of (reg{i}), which is ＿, and now I will increment
        this register by 1."
     ii) Increment reg{i} by 1.
        [Intermediate answer]:  "I incremented (reg{i}) by 1, and the
        updated value is ＿, as I added 1 to its original value."
    iii) Update pc by adding 1.
        [Intermediate answer]:  "I updated the value of (pc) by adding
        1, and now pc is ＿"
  b) If the instruction is of the form B(reg{i}, new_pc):
     i) Read the value of reg{i} from the <registers> sections in
       memory.
       [Intermediate answer]:  "Since the instruction is of the form
       B(reg{i}), I checked the current value of (reg{i}), which is ＿,
       and this will determine our next action."
    ii) If reg{i} equals 0, set pc to new_pc in <program counter>.
       [Intermediate answer]:  "Since (reg{i}) is 0, I updated the
       value of pc to (new_pc), which is ＿." iii) Otherwise, I
       decrement reg{i} by 1 and increment pc by 1.
       [Intermediate answer]:  "Since (reg{i}) is not 0, I decremented
       (reg{i}) by 1 to get the updated value of ＿, and incremented pc
       to ＿"

4) Output the new memory (all <program counter>, <registers>, and
<instructions>), after making the updates, within <memory></memory>
tags.
[Final answer]:  "After executing the instruction, the final state of
<memory> is:"

Execute steps 1) to 4) once.  Replace all placeholders '＿', and all the
[Intermediate answer] tags with the appropriate intermediate responses.
In step 3), output only the corresponding intermediate answer for the
chosen operation (3.a or 3.b).
In step 4), output the entire memory within <memory></memory> tags, as
the [Final answer].
Answer:

### A.1.2  SUBLEQ OISC

**Direct output prompting**

Below are the steps to execute the current instruction A in <memory>:

STEPS:
1) Read the value of pc from the <program counter> section in memory.

2) Read the instruction at line{pc} from the <instructions> section in
memory.

3) Since the instruction is of the form A(reg{i}, reg{j}, new_pc), execute
the following:

   i) Read the value of reg{i} from the <registers> sections in memory.
  ii) Read the value of reg{j} from the <registers> sections in memory.
 iii) Do the calculation reg{j} = reg{j} - reg{i}.

4) Update the program counter (pc).  Execute only one of the following
operations, based on the updated value of reg{j}:

   i) If the updated value of reg{j} is <= 0, set pc to new_pc.

   ii) Otherwise, if the updated value of reg{j} is > 0, set pc to pc +
       1.

5) Output the updated memory (all <program counter>, <registers>, and
<instructions>), after making the updates, within <memory></memory>
tags.  [Final answer]:  "After executing the instruction A, the final
state of <memory> is:"

Execute steps 1) to 5) once.  Output only the updated memory, as
described in step 5) within <memory></memory> tags, as the Answer.
Answer:

## Chain-of-thought prompting

Below are the steps to execute the current instruction A in <memory>:

STEPS:
1) Read the value of pc from the <program counter> section in memory.
[Intermediate answer]:  "Searching in the <program counter> section, I
found that the current value of pc is ＿, which points to the instruction
we need to execute."

2) Read the instruction at line{pc} from the <instructions> section in
memory.
[Intermediate answer]:  "Searching in the <instructions> section, I
located the instruction at line{pc}, which is A(reg{i}, reg{j}, new＿pc),
and this specifies the operation to be performed on the registers."

3) Since the instruction is of the form A(reg{i}, reg{j}, new＿pc), execute
the following:

    i) Read the value of reg{i} from the <registers> sections in memory.
       [Intermediate answer]:  "I searched in <registers> section and
       retrieved the current value of (reg{i}) which is ＿."
   ii) Read the value of reg{j} from the <registers> sections in memory.
       [Intermediate answer]:  "I searched in <registers> section and
       retrieved the current value of (reg{j}) which is ＿."
  iii) Do the calculation reg{j} = reg{j} – reg{i}.
       [Intermediate answer]:  "I did the calculation (reg{j}) = (reg{j})
       – (reg{i}), and the updated value of (reg{j}) is (＿) – (＿) = (＿)."

4) Update the program counter (pc).  Execute only one of the following
operations, based on the updated value of reg{j}:

    i) If the new updated of reg{j} is <= 0, set pc to new＿pc.
       [Intermediate answer]:  "Since the updated value of (reg{j}) is ＿
       which is <= 0, I updated the value of (pc) to (new＿pc), which is
       ＿."
   ii) Otherwise, if the updated value of reg{j} is > 0, set pc to pc +
       1.
       [Intermediate answer]:  "Since the updated value of (reg{j}) is ＿
       which is > 0, I updated the value of (pc) to (pc + 1), which is
       ＿."

5) Output the updated memory (all <program counter>, <registers>, and
<instructions>), after making the updates, within <memory></memory>
tags.
[Final answer]:  "After executing the instruction A, the updated state of
<memory> is:"

Execute steps 1) to 5) once.  Replace all placeholders '＿', and all the
[Intermediate answer] tags with the appropriate intermediate responses.
In step 4), output only the corresponding intermediate answer for the
chosen operation (4.i or 4.ii).
In step 5), output the entire memory within <memory></memory> tags, as

```
the [Final answer].
Answer:
```

## A.2   MULTIPLE-INFERENCES PER INSTRUCTION

Below, we present the detailed prompts that were used in the *multiple-inferences per instruction* experiments. Recall that in this approach, the model is invoked 3 times for reading pc, fetching line{pc} and executing line{pc}, respectively.

### A.2.1   MINSKY MACHINE

**Read** pc

```
[Task]:  Read pc and output only its value from the <program counter>
section in memory
[Answer]:  value of pc=
```

**Fetch** line{pc} **[given** pc **]**

```
[Task]:  Read and output the instruction at line{pc} from the
<instructions> section in memory
[Answer]:  line{pc}=
```

**Execute** line{pc} **[given** pc **, and** line{pc}**]**

```
[Task]:
The current instruction is {instruction}, and the current value of pc is
{pc}.  Below are the steps to execute {instruction} in <memory>:
STEPS:
1) Based on the instruction {instruction} and the current value pc={pc},
choose only one of the steps a) or b) (and not both).  Explain why, and
execute one of the following operations:

    a) If the instruction is of the form A(reg{i}):

        i) Read the value of reg{i} from the <registers> sections in
           memory.
           [Intermediate answer]:  "Since the instruction is of the form
           A(reg{i}), I searched in <registers> section and retrieved the
           current value of (reg{i}), which is __, and now I will increment
           this register by 1."
       ii) Increment reg{i} by 1.
           [Intermediate answer]:  "I incremented (reg{i}) by 1, and the
           updated value is __, as I added 1 to its original value."
      iii) Update pc by adding 1.
           [Intermediate answer]:  "I updated the value of (pc) by adding
           1, and now pc is __"
    b) If the instruction is of the form B(reg{i}, new_pc):

        i) Read the value of reg{i} from the <registers> sections in
           memory.
           [Intermediate answer]:  "Since the instruction is of the form
           B(reg{i}), I checked the current value of (reg{i}), which is __,
           and this will determine our next action."
       ii) If reg{i} equals 0, set pc to new_pc in <program counter>.
           [Intermediate answer]:  "Since (reg{i}) is 0, I updated the
           value of pc to (new_pc), which is __."
      iii) Otherwise, I decrement reg{i} by 1 and increment pc by 1.
           [Intermediate answer]:  "Since (reg{i}) is not 0, I decremented
           (reg{i}) by 1 to get the updated value of __, and incremented pc
           to __"

2) Output the new memory (all <program counter>, <registers>, and
<instructions>), after making the updates, within <memory></memory>
```

```
tags.
[Final answer]:  "After executing the instruction, the final state of
<memory> is:"

Execute steps 1) and 2).  Replace all placeholders '__', and all the
[Intermediate answer] tags with the appropriate intermediate responses.
In step 1), output only the corresponding intermediate answers for the
chosen operations (1.a or 1.b).
In step 2), output the entire memory within <memory></memory> tags, as
the [Final answer].
[Answer]:
```

### A.2.2 SUBLEQ OISC

**Read** pc

```
[Task]:  Read pc and output only its value from the <program counter>
section in memory
[Answer]:  value of pc=
```

**Fetch** line{pc} **[given** pc **]**

```
[Task]:  Read and output the instruction at line{pc} from the
<instructions> section in memory
[Answer]:  line{pc}=
```

**Execute** line{pc} **[given** pc **, and** line{pc}**]**

```
[Task]:
The current instruction is {instruction}, and the current value of pc is
{pc}.  Below are the steps to execute {instruction} in <memory>:
STEPS:
1) Since the instruction is of the form A(reg{i}, reg{j}, new_pc), execute
the following:

    i) Read the value of reg{i} from the <registers> sections in memory.
       [Intermediate answer]:  "I searched in <registers> section and
       retrieved the current value of (reg{__}) which is __."
   ii) Read the value of reg{j} from the <registers> sections in memory.
       [Intermediate answer]:  "I searched in <registers> section and
       retrieved the current value of (reg{__}) which is __."
  iii) Do the calculation reg{j} = reg{j} - reg{i}.  [Intermediate
       answer]:  "I did the calculation (reg{__}) = (reg{__}) - (reg{__}),
       and the updated value of (reg{__}) is (__) - (__) = (__)."

2) Update the program counter (pc), whose current value is {pc}.  Execute
only one of the following operations, based on the updated value of
reg{j}:

    i) If the updated value of reg{j} is <= 0, set pc to new_pc.
       [Intermediate answer]:  "Since the updated value of (reg{__}) is __
       which is <= 0, I updated the value of (pc) to (new_pc), which is
       __."
   ii) Otherwise, if the updated value of reg{j} is > 0, set pc to pc +
       1.
       [Intermediate answer]:  "Since the updated value of (reg{__}) is __
       which is > 0, I updated the value of (pc) to (pc + 1), which is
       __."

3) Output the updated memory (all <program counter>, <registers>, and
<instructions>), after making the updates, within <memory></memory>
tags.
```

```
[Final answer]:  "After executing the instruction A, the updated state of
<memory> is:"

Execute steps 1) and 2).  Replace all placeholders '_', and all the
[Intermediate answer] tags with the appropriate intermediate responses.
In step 2), output only the corresponding intermediate answers for the
chosen operations (2.i or 2.ii).
In step 3), output the entire memory within <memory></memory> tags, as
the [Final answer].
[Answer]:
```

