# OpenReview forum: "Can General-Purpose Language Models Emulate a General-Purpose Computer In-Context?"
_ICLR.cc/2024/Conference — ICLR 2024 Conference Withdrawn Submission_

### Official Review · Reviewer_ZKpH · 2023-10-30

**Soundness:** 3 good
**Presentation:** 3 good
**Contribution:** 1 poor
**Rating:** 3
**Confidence:** 4

**Summary:**

The paper studies if pretrained LLMs could emulate a memory-bounded, reduced instruction-set based computer by executing random programs through looped inference calls. The work examines Minsky Machine and SUBLEQ OISC, two basic computer models with Turing completeness. Various GPT and Claude LLMs have been tested on the task execution accuracy, i.e., if the LLM could correctly process programs and matches results with the actual machines. The results show that GPT-4 is the most performance with others faring worse.

**Strengths:**

+ Detailed description of the task setup, the inference approaches, and prompting methods, with examples presented in the appendix.
+ Sufficient experiments with varying level of randomness to show how different models perform and detailed comparison among them.

**Weaknesses:**

In general, I feel pathetic towards this line of research, comparing a STOCHASTIC and UNINTERPRETABLE model with DETERMINISTIC, INTERPRETABLE, and even THEORETICALLY SOUND model. Turing machines are deterministic computation mechanisms and the entire computer empire to date is built around it, as is THE Ground-Truth model in the "model-based" approach. However, LLMs are purely statistically, a completely model-free approach. Whenever you have an extremely reliable Ground-Truth model-based model, why would you rely on a stochastic model-free model that is, even worse, uninterpretable and theoretically unknown? I would appreciate it if a task has unknown mechanism and you use statistically methods to approximate it. But for tasks considered in this work, easily and reliably solved with these machines, why would you throw a LLM at them?

From another perspective, even if your LLM, no matter which, achieves 100% accuracy on your dataset, would you be actually using it for your computer program tasks? No, because it is only reliable on your selected dataset and not guaranteed in the wild. However, theoretical computers are provably reliable in the wild. When you have the choice of the two, why would someone pick the LLM when facing real-world problems that can be solved with computers?

What is the take-away from this work except the already known ``scaling is powerful''? GPT-4's superior capability has been shown in a variety of tasks, and adding this one doesn't tell us what parts in GPT-4 make it much better than others in the problem, except the magnitude of data used.

With these high-level questions unanswered, I fail to see the contribution of this work.

**Questions:**

See weaknesses above.

---

### Official Review · Reviewer_xPeX · 2023-10-30

**Soundness:** 4 excellent
**Presentation:** 4 excellent
**Contribution:** 1 poor
**Rating:** 3
**Confidence:** 5

**Summary:**

This paper benchmarks LLMs on the task of following instructions that cause the model to emulate a computation model. The results show that models generally do this task quite well, especially if prompted appropriately.

**Strengths:**

1. The paper is generally clearly written, it is self-contained and easy to follow.
2. The paper offers a comprehensive study of instruction following in the context of emulating simple computational models.

**Weaknesses:**

1. The authors set the aim to evaluate the capabilities of an LLM to emulate a general-purpose computer “in-context” and “without the use of external mechanisms”. However, they do resort to an external mechanism: the parser and the implicit “looper” that they do not discuss. Similarly, because of the “looper”, this is not really “in-context” emulation. The paper does not at all evaluate how adept these models are at truly in-context evaluation. That is: multi-step execution where the result of each step is added to the context and the model iterates autoregressively over the memory states without any external mechanism, incl. a parser.
2. In its current form, the paper essentially benchmarks for instruction following abilities but only for a couple of parameterised instructions (the prompts in App. A) with various values for <memory>. I am not convinced that this is the same as “emulating a general-purpose computer in-context”.
3. The paper claims “The emergence of this skill, despite not being part of the training objective, has noteworthy implications”. First, it is not clear whether that is emergence because it is not known what training data the models have seen. And, as mentioned above, what the experiments test is simple instruction-following which at least some of these models have been fine-tuned for. Even if we assume that this is emergence, though, these implications are never properly addressed. How come a language model that can follow instructions to emulate CPU instructions is more dangerous than, e.g., a standard Python interpreter?
4. The abstract says “we conduct studies to examine failure modes related to the emulated computer functionality” but I could not find a discussion of these failure modes.

**Questions:**

1. Can you provide an evaluation that is “in-context” and “without the use of external mechanisms”?
2. Can you explain what studies on failure modes were performed and what the respective results are?
3. How does this work differ from other instruction-following evaluations and benchmarks?

Typos:
- Pg. 4: I assume you meant to say “This means that the results are NOT direct memorization…” instead of “This means that the results are direct memorization…”

---

### Official Review · Reviewer_Brom · 2023-10-31

**Soundness:** 1 poor
**Presentation:** 1 poor
**Contribution:** 1 poor
**Rating:** 3
**Confidence:** 5

**Summary:**

This paper studies whether a pre-trained LLM can emulate a memory-bounded reduced instruction-set-based computer by executing random programs through looped inference calls. It studies this without the aid of an external memory mechanism, and studies two computational models of SUBLEQ One-Instruction Set Computer (OISC) and the Minksy counter machine. The findings presented show that certain LLMs are capable of executing general-purpose instructions of these computational models.

**Strengths:**

* The paper is clearly written
* It is promising to see multiple LLMs, and ablations of varying the number of registers, instructions and the prompting technique used.

**Weaknesses:**

* The work of Schuurmans 2023 already proves that transformer-based LLMs can exactly simulate (emulate) a Universal Turing Machine with an associative read-write memory. This paper studies the question of *"Can pretrained LLMs emulate, in-context, a general-purpose computer, without the use of external mechanisms (such as memory or interpreters)?"*; however, this question is a subset of the work of Schuurmans 2023, and therefore this paper's proposed core hypothesis and contribution is not novel.
  * **Reducing the associative external memory to an in-context memory is impractical**. Replacing a simple associative external memory mechanism such as a dictionary of a key value mapping as in Schuurmans 2023, to that of the in-context memory proposed in this paper leads to issues of:
    * **(1)** *Significantly less scalable, as the entire key-value memory has to explicitly included within the context*, which leads to ineffective use of the constrained context window tokens, compared to using a possibly unbounded external key-value dictionary.
    * **(2)** *The reliability of relying the LLM to write and read to the key-value memory introduces errors* (as shown in Figure 5 and Figure 7), whereas relying on a built-in key-value map of a simple Python dictionary (Schuurmans 2023) introduces no errors.
    * **(3)** *The context acts as an external small finite memory store*. As the entire memory is included in context, and after each LLM response it is updated by *"a simple regular expression designed to identify the portion of the response enclosed within the `<memory></memory>` tags."*
  * As the context can be viewed as an external small finite memory store, the hypothesis then becomes *"Can pretrained LLMs emulate, in-context, a general-purpose computer?"*; however, again, Schuurmans 2023 has already proved that LLMs with an external memory can exactly simulate a Universal Turing Machine; which is equivalent to emulating a general-purpose computer. Although the two proposed general-purpose computational models of SUBLEQ OISC and the Minsky counter machine are both Turing complete and the Minsky counter machine is only a Universal Turing Machine for a large enough number of instructions, i.e., 37 instructions (Gregušová et al. 1979). Schuurmans 2023 by proving an LLM with external memory can simulate a Universal Turing Machine (UTM), means that an LLM with external memory can simulate other Turing machines. Naturally a UTM is Turing complete; where being Turing complete means that it compute any computable sequence (Turing, 1936). Therefore the claim of *"Can pretrained LLMs emulate, in-context, a general-purpose computer?"* has already been claimed by Schuurmans 2023, and this paper's core contribution, study and results are not novel.
* Methodology to verify simulation is lacking explanation. The paper generates "random sets of instructions", which could be suitable when the input and output combination space is too large to be exhaustively verified for each state. However, existing work uses a more comprehensive approach to verify the simulation of a general-purpose computer, such as Schuurmans 2023, that proves by exhaustively verifying all possible input and output combinations of a simple UTM.
* Lacking experimental details. I could not find any details about how the error bars are computer in Figure 1, Figure 5, Figure 6 and Figure 7.
* Could compare against more prompting techniques, e.g., Plan-and-Solve (Wang et al. 2023).
* The statement of *"In all cases, we observe that the models achieve near-perfect accuracy, which appears to be unaffected by the increase in the number of registers and instructions in the memory state."*, the empirical evidence given is only for a small amount of instructions still up to 30, and only for maximally 20 registers. It could be informative to the reader to see what happens when these values are set to their maximum, as defined by filling up the entire context window---I would expect a degradation at larger numbers (Liu, 2023).




Typos:

* Abstract: "All this within the" -> "All within the"
* Page 2, "absent external constraints" -> "absent of external constraints"
* Page 2, "moderm" -> "modern"
* Page 4, "This means that the results are direct memorization of the specific data used during training" -> "This means that the results are **not** direct memorization of the specific data used during training".
* Page 6, "In this series" -> "In these series"
* Inconsistent paragraph formatting, Page 2-3, bold paragraph heading has a full stop, later paragraph headings are missing the full stop. It could be useful to be consistent with one convention throughout the paper.



---

References:

* Schuurmans, Dale. "Memory augmented large language models are computationally universal." arXiv preprint arXiv:2301.04589 (2023).
* Gregušová, Ľudmila, and Ivan Korec. "Small universal Minsky machines." International Symposium on Mathematical Foundations of Computer Science. Berlin, Heidelberg: Springer Berlin Heidelberg, 1979.
* Turing, Alan Mathison. "On computable numbers, with an application to the Entscheidungsproblem." J. of Math 58.345-363 (1936): 5.
* Wang, Lei, et al. "Plan-and-solve prompting: Improving zero-shot chain-of-thought reasoning by large language models." arXiv preprint arXiv:2305.04091 (2023).
* Liu, Nelson F., et al. "Lost in the middle: How language models use long contexts." arXiv preprint arXiv:2307.03172 (2023).

**Questions:**

* See all weaknesses
* Can you run further ablations to increase the number of instructions and registers used to their respective maximum values that will fit within the context?

---

### Official Review · Reviewer_nL65 · 2023-10-31

**Soundness:** 2 fair
**Presentation:** 3 good
**Contribution:** 1 poor
**Rating:** 3
**Confidence:** 4

**Summary:**

This paper explores the ability of pretrained language models (PLMs) to simulate Turing machines in context, i.e., without access to an external memory. The work uses two simple Turing complete languages: the SUBLEQ and the Minksy machine. The results show that several commercial PLMs are able to read and write registers, as well as fetch instructions with near perfect zero-shot accuracy. Furthermore, PLMs can be induced to execute accurately using prompting techniques (such as chain-of-thought). The work concludes that PLMs have acquired an emergent ability to model general-purpose computation.

**Strengths:**

The idea to explore LMs through the lens of Turing completeness is interesting, and could reveal a number of insights about the capabilities and limitations of LMs. Though a number of works have looked at the capabilities of LMs in the context of programs (as well as theoretical work showing that Transformers augmented with external memory are formally Turing complete), I'm not aware of any empirical works which have explored this from an empirical angle.

The writing and experimental set up are generally clear. This research direction could have wide-ranging, significant implications for the future of LLMs as well.

**Weaknesses:**

I have a major reservation about the work in its current iteration: the authors position the significance of the results as **emergent abilities**; however, I don't see how this claim can be made for the closed-source models they use in the experiments. For instance, I think the significance of the paper would be close to nil if we assume that the models been trained explicitly on data containing SUBLEQ programs and their execution (i.e., I don't think it's an interesting question whether one could *supervise* a Transformer to execute SUBLEQ programs). Since we cannot rule out this possibility (certainly, at least 2 of the models used in the experiments can recite fairly detailed descriptions of SUBLEQ and Minksy machine languages), the main claims of the paper (about emergent abilities) are not supported by the experiments.

A lesser concern is that Turing machines need access to an infinite tape by definition, so there is a bit of a disconnect when studying the ability of LMs to emulate Turing machines in a finite context. I don't think this is a fatal flaw, but I do think this warrants a discussion.

**Questions:**

Can you clarify the significance of the results, given that an LM cannot actually emulate a full Turing machine in finite context?

Do you have any conviction that these closed source models were not trained on data containing SUBLEQ (or Minksy machine) programs and their executions? Are the results still significant if they were trained on such data?

Have you tried to reproduce the results for open source models, or performed a training data contamination study?